# Learning the Number of Neurons in Deep Networks

**Jose M. Alvarez**[*]
Data61 @ CSIRO
Canberra, ACT 2601, Australia
jose.alvarez@data61.csiro.au

**Mathieu Salzmann**
CVLab, EPFL
CH-1015 Lausanne, Switzerland
mathieu.salzmann@epfl.ch

## Abstract

Nowadays, the number of layers and of neurons in each layer of a deep network are typically set manually. While very deep and wide networks have proven effective in general, they come at a high memory and computation cost, thus making them impractical for constrained platforms. These networks, however, are known to have many redundant parameters, and could thus, in principle, be replaced by more compact architectures. In this paper, we introduce an approach to automatically determining the number of neurons in each layer of a deep network during learning. To this end, we propose to make use of a group sparsity regularizer on the parameters of the network, where each group is defined to act on a single neuron. Starting from an overcomplete network, we show that our approach can reduce the number of parameters by up to 80% while retaining or even improving the network accuracy.

## 1 Introduction

Thanks to the growing availability of large-scale datasets and computation power, Deep Learning has recently generated a quasi-revolution in many fields, such as Computer Vision and Natural Language Processing. Despite this progress, designing a deep architecture for a new task essentially remains a dark art. It involves defining the number of layers and of neurons in each layer, which, together, determine the number of parameters, or complexity, of the model, and which are typically set manually by trial and error.

A recent trend to avoid this issue consists of building very deep [Simonyan and Zisserman, 2014] or ultra deep [He et al., 2015] networks, which have proven more expressive. This, however, comes at a significant cost in terms of memory requirement and speed, which may prevent the deployment of such networks on constrained platforms at test time and complicate the learning process due to exploding or vanishing gradients.

Automatic model selection has nonetheless been studied in the past, using both constructive and destructive approaches. Starting from a shallow architecture, constructive methods work by incrementally incorporating additional parameters [Bello, 1992] or, more recently, layers to the network [Simonyan and Zisserman, 2014]. The main drawback of this approach stems from the fact that shallow networks are less expressive than deep ones, and may thus provide poor initialization when adding new layers. By contrast, destructive techniques exploit the fact that very deep models include a significant number of redundant parameters [Denil et al., 2013, Cheng et al., 2015], and thus, given an initial deep network, aim at reducing it while keeping its representation power. Originally, this has been achieved by removing the parameters [LeCun et al., 1990, Hassibi et al., 1993] or the neurons [Mozer and Smolensky, 1988, Ji et al., 1990, Reed, 1993] that have little influence on the output. While effective this requires analyzing every parameter/neuron independently, e.g., via the network Hessian, and thus does not scale well to large architectures. Therefore, recent trends

---

[*]http://www.josemalvarez.net.

to performing network reduction have focused on training shallow or thin networks to mimic the behavior of large, deep ones [Hinton et al., 2014, Romero et al., 2015]. This approach, however, acts as a post-processing step, and thus requires being able to successfully train an initial deep network.

In this paper, we introduce an approach to automatically selecting the number of neurons in each layer of a deep architecture simultaneously as we learn the network. Specifically, our method does not require training an initial network as a pre-processing step. Instead, we introduce a group sparsity regularizer on the parameters of the network, where each group is defined to act on the parameters of one neuron. Setting these parameters to zero therefore amounts to canceling the influence of a particular neuron and thus removing it entirely. As a consequence, our approach does not depend on the success of learning a redundant network to later reduce its parameters, but instead jointly learns the number of relevant neurons in each layer and the parameters of these neurons.

We demonstrate the effectiveness of our approach on several network architectures and using several image recognition datasets. Our experiments demonstrate that our method can reduce the number of parameters by up to 80% compared to the complete network. Furthermore, this reduction comes at no loss in recognition accuracy; it even typically yields an improvement over the complete network. In short, our approach not only lets us automatically perform model selection, but it also yields networks that, at test time, are more effective, faster and require less memory.

## 2 Related work

Model selection for deep architectures, or more precisely determining the best number of parameters, such as the number of layers and of neurons in each layer, has not yet been widely studied. Currently, this is mostly achieved by manually tuning these hyper-parameters using validation data, or by relying on very deep networks [Simonyan and Zisserman, 2014, He et al., 2015], which have proven effective in many scenarios. These large networks, however, come at the cost of high memory footprint and low speed at test time. Furthermore, it is well-known that most of the parameters in such networks are redundant [Denil et al., 2013, Cheng et al., 2015], and thus more compact architectures could do as good a job as the very deep ones.

While sparse, some literature on model selection for deep learning nonetheless exists. In particular, a forerunner approach was presented in [Ash, 1989] to dynamically add nodes to an existing architecture. Similarly, [Bello, 1992] introduced a constructive method that incrementally grows a network by adding new neurons. More recently, a similar constructive strategy was successfully employed by [Simonyan and Zisserman, 2014], where their final very deep network was built by adding new layers to an initial shallower architecture. The constructive approach, however, has a drawback: Shallow networks are known not to handle non-linearities as effectively as deeper ones [Montufar et al., 2014]. Therefore, the initial, shallow architectures may easily get trapped in bad optima, and thus provide poor initialization for the constructive steps.

In contrast with constructive methods, destructive approaches to model selection start with an initial deep network, and aim at reducing it while keeping its behavior unchanged. This trend was started by [LeCun et al., 1990, Hassibi et al., 1993] to cancel out individual parameters, and by [Mozer and Smolensky, 1988, Ji et al., 1990, Reed, 1993], and more recently [Liu et al., 2015], when it comes to removing entire neurons. The core idea of these methods consists of studying the saliency of individual parameters or neurons and remove those that have little influence on the output of the network. Analyzing individual parameters/neurons, however, quickly becomes computationally expensive for large networks, particularly when the procedure involves computing the network Hessian and is repeated multiple times over the learning process. As a consequence, these techniques have no longer been pursued in the current large-scale era. Instead, the more recent take on the destructive approach consists of learning a shallower or thinner network that mimics the behavior of an initial deep one [Hinton et al., 2014, Romero et al., 2015], which ultimately also reduces the number of parameters of the initial network. The main motivation of these works, however, was not truly model selection, but rather building a more compact network.

As a matter of fact, designing compact models also is an active research focus in deep learning. In particular, in the context of Convolutional Neural Networks (CNNs), several works have proposed to decompose the filters of a pre-trained network into low-rank filters, thus reducing the number of parameters [Jaderberg et al., 2014b, Denton et al., 2014, Gong et al., 2014]. However, this approach, similarly to some destructive methods mentioned above, acts as a post-processing step, and

thus requires being able to successfully train an initial deep network. Note that, in a more general context, it has been shown that a two-step procedure is typically outperformed by one-step, direct training [Srivastava et al., 2015]. Such a direct approach has been employed by [Weigend et al., 1991] and [Collins and Kohli, 2014] who have developed regularizers that favor eliminating some of the parameters of the network, thus leading to lower memory requirement. The regularizers are minimized simultaneously as the network is learned, and thus no pre-training is required. However, they act on individual parameters. Therefore, similarly to [Jaderberg et al., 2014b, Denton et al., 2014] and to other parameter regularization techniques [Krogh and Hertz, 1992, Bartlett, 1996], these methods do not perform model selection; the number of layers and neurons per layer is determined manually and won't be affected by learning.

By contrast, in this paper, we introduce an approach to automatically determine the number of neurons in each layer of a deep network. To this end, we design a regularizer-based formulation and therefore do not rely on any pre-training. In other words, our approach performs model selection and produces a compact network in a single, coherent learning framework. To the best of our knowledge, only three works have studied similar group sparsity regularizers for deep networks. However, [Zhou et al., 2016] focuses on the last fully-connected layer to obtain a compact model, and [S. Tu, 2014] and [Murray and Chiang, 2015] only considered small networks. Our approach scales to datasets and architectures two orders of magnitude larger than in these last two works with minimum (and tractable) training overhead. Furthermore, these three methods define a single global regularizer. By contrast, we work in a per-layer fashion, which we found more effective to reduce the number of neurons by large factors without accuracy drop.

## 3   Deep Model Selection

We now introduce our approach to automatically determining the number of neurons in each layer of a deep network while learning the network parameters. To this end, we describe our framework for a general deep network, and discuss specific architectures in the experiments section.

A general deep network can be described as a succession of $L$ layers performing linear operations on their input, intertwined with non-linearities, such as Rectified Linear Units (ReLU) or sigmoids, and, potentially, pooling operations. Each layer $l$ consists of $N_l$ neurons, each of which is encoded by parameters $\theta_l^n = [\mathbf{w}_l^n, b_l^n]$, where $\mathbf{w}_l^n$ is a linear operator acting on the layer's input and $b_l^n$ is a bias. Altogether, these parameters form the parameter set $\Theta = \{\theta_l\}_{1 \leq l \leq L}$, with $\theta_l = \{\theta_l^n\}_{1 \leq n \leq N_l}$. Given an input signal $\mathbf{x}$, such as an image, the output of the network can be written as $\hat{y} = f(\mathbf{x}, \Theta)$, where $f(\cdot)$ encodes the succession of linear, non-linear and pooling operations.

Given a training set consisting of $N$ input-output pairs $\{(\mathbf{x}_i, y_i)\}_{1 \leq i \leq N}$, learning the parameters of the network can be expressed as solving an optimization of the form

$$\min_{\Theta} \frac{1}{N} \sum_{i=1}^{N} \ell(y_i, f(\mathbf{x}_i, \Theta)) + r(\Theta) \;, \tag{1}$$

where $\ell(\cdot)$ is a loss that compares the network prediction with the ground-truth output, such as the logistic loss for classification or the square loss for regression, and $r(\cdot)$ is a regularizer acting on the network parameters. Popular choices for such a regularizer include weight-decay, i.e., $r(\cdot)$ is the (squared) $\ell_2$-norm, of sparsity-inducing norms, e.g., the $\ell_1$-norm.

Recall that our goal here is to automatically determine the number of neurons in each layer of the network. We propose to do this by starting from an overcomplete network and canceling the influence of some of its neurons. Note that none of the standard regularizers mentioned above achieve this goal: The former favors small parameter values, and the latter tends to cancel out individual parameters, but not complete neurons. In fact, a neuron is encoded by a group of parameters, and our goal therefore translates to making entire groups go to zero. To achieve this, we make use of the notion of group sparsity [Yuan and Lin., 2007]. In particular, we write our regularizer as

$$r(\Theta) = \sum_{l=1}^{L} \lambda_l \sqrt{P_l} \sum_{n=1}^{N_l} \|\theta_l^n\|_2 \;, \tag{2}$$

where, without loss of generality, we assume that the parameters of each neuron in layer $l$ are grouped in a vector of size $P_l$, and where $\lambda_l$ sets the influence of the penalty. Note that, in the general case,

this weight can be different for each layer $l$. In practice, however, we found most effective to have two different weights: a relatively small one for the first few layers, and a larger weight for the remaining ones. This effectively prevents killing too many neurons in the first few layers, and thus retains enough information for the remaining ones.

While group sparsity lets us effectively remove some of the neurons, exploiting standard regularizers on the individual parameters has proven effective in the past for generalization purpose [Bartlett, 1996, Krogh and Hertz, 1992, Theodoridis, 2015, Collins and Kohli, 2014]. To further leverage this idea within our automatic model selection approach, we propose to exploit the sparse group Lasso idea of [Simon et al., 2013]. This lets us write our regularizer as

$$r(\Theta) = \sum_{l=1}^{L} \left( (1-\alpha)\lambda_l \sqrt{P_l} \sum_{n=1}^{N_l} \|\theta_l^n\|_2 + \alpha\lambda_l \|\theta_l\|_1 \right) , \tag{3}$$

where $\alpha \in [0,1]$ sets the relative influence of both terms. Note that $\alpha = 0$ brings us back to the regularizer of Eq. 2. In practice, we experimented with both $\alpha = 0$ and $\alpha = 0.5$.

To solve Problem (1) with the regularizer defined by either Eq. 2 or Eq. 3, we follow a proximal gradient descent approach [Parikh and Boyd, 2014]. In our context, proximal gradient descent can be thought of as iteratively taking a gradient step of size $t$ with respect to the loss $\sum_{i=1}^{N} \ell(y_i, f(\mathbf{x}_i, \Theta))$ only, and, from the resulting solution, applying the proximal operator of the regularizer. In our case, since the groups are non-overlapping, we can apply the proximal operator to each group independently. Specifically, for a single group, this translates to updating the parameters as

$$\tilde{\theta}_l^n = \operatorname*{argmin}_{\theta_l^n} \frac{1}{2t} \|\theta_l^n - \hat{\theta}_l^n\|_2^2 + r(\Theta) , \tag{4}$$

where $\hat{\theta}_l^n$ is the solution obtained from the loss-based gradient step. Following the derivations of [Simon et al., 2013], and focusing on the regularizer of Eq. 3 of which Eq. 2 is a special case, this problem has a closed-form solution given by

$$\tilde{\theta}_l^n = \left( 1 - \frac{t(1-\alpha)\lambda_l\sqrt{P_l}}{\|S(\hat{\theta}_l^n, t\alpha\lambda_l)\|_2)} \right)_+ S(\hat{\theta}_l^n, t\alpha\lambda_l) , \tag{5}$$

where $+$ corresponds to taking the maximum between the argument and 0, and $S(\cdot)$ is the soft-thresholding operator defined elementwise as

$$(S(\mathbf{z}, \tau))_j = \operatorname{sign}(\mathbf{z}_j)(|\mathbf{z}_j| - \tau)_+ . \tag{6}$$

The learning algorithm therefore proceeds by iteratively taking a gradient step based on the loss only, and updating the variables of all the groups according to Eq. 5. In practice, we follow a stochastic gradient descent approach and work with mini-batches. In this setting, we apply the proximal operator at the end of each epoch and run the algorithm for a fixed number of epochs.

When learning terminates, the parameters of some of the neurons will have gone to zero. We can thus remove these neurons entirely, since they have no effect on the output. Furthermore, when considering fully-connected layers, the neurons acting on the output of zeroed-out neurons of the previous layer also become useless, and can thus be removed. Ultimately, removing all these neurons yields a more compact architecture than the original, overcomplete one.

## 4  Experiments

In this section, we demonstrate the ability of our method to automatically determine the number of neurons on the task of large-scale classification. To this end, we study three different architectures and analyze the behavior of our method on three different datasets, with a particular focus on parameter reduction. Below, we first describe our experimental setup and then discuss our results.

### 4.1  Experimental setup

**Datasets:** For our experiments, we used two large-scale image classification datasets, ImageNet [Russakovsky et al., 2015] and Places2-401 [Zhou et al., 2015]. Furthermore, we conducted

additional experiments on the character recognition dataset of [Jaderberg et al., 2014a]. ImageNet contains over 15 million labeled images split into 22,000 categories. We used the ILSVRC-2012 [Russakovsky et al., 2015] subset consisting of 1000 categories, with 1.2 million training images and 50,000 validation images. Places2-401 [Zhou et al., 2015] is a large-scale dataset specifically created for high-level visual understanding tasks. It consists of more than 10 million images with 401 unique scene categories. The training set comprises between 5,000 and 30,000 images per category. Finally, the ICDAR character recognition dataset of [Jaderberg et al., 2014a] consists of 185,639 training and 5,198 test samples split into 36 categories. The training samples depict characters collected from text observed in a number of scenes and from synthesized datasets, while the test set comes from the ICDAR2003 training set after removing all non-alphanumeric characters.

**Architectures:** For ImageNet and Places2-401, our architectures are based on the VGG-B network (BNet) [Simonyan and Zisserman, 2014] and on DecomposeMe$_8$ (Dec$_8$) [Alvarez and Petersson, 2016]. **BNet** consists of 10 convolutional layers followed by three fully-connected layers. In our experiments, we removed the first two fully-connected layers. As will be shown in our results, while this reduces the number of parameters, it maintains the accuracy of the original network. Below, we refer to this modified architecture as **BNet**$^C$. Following the idea of low-rank filters, **Dec**$_8$ consists of 16 convolutional layers with 1D kernels, effectively modeling 8 2D convolutional layers. For ICDAR, we used an architecture similar to the one of [Jaderberg et al., 2014b]. The original architecture consists of three convolutional layers with a maxout layer [Goodfellow et al., 2013] after each convolution, followed by one fully-connected layer. [Jaderberg et al., 2014b] first trained this network and then decomposed each 2D convolution into 2 1D kernels. Here, instead, we directly start with 6 1D convolutional layers. Furthermore, we replaced the maxout layers with max-pooling. As shown below, this architecture, referred to as **Dec**$_3$, yields similar results as the original one, referred to as **MaxOut**.

**Implementation details:** For the comparison to be fair, all models including the baselines were trained from scratch on the same computer using the same random seed and the same framework. More specifically, for ImageNet and Places2-401, we used the torch-7 multi-gpu framework [Collobert et al., 2011] on a Dual Xeon 8-core E5-2650 with 128GB of RAM using three Kepler Tesla K20m GPUs in parallel. All models were trained for a total of 55 epochs with 12,000 batches per epoch and a batch size of 48 and 180 for BNet and Dec$_8$, respectively. These variations in batch size were mainly due to the memory and runtime limitations of BNet. The learning rate was set to an initial value of 0.01 and then multiplied by 0.1. Data augmentation was done through random crops and random horizontal flips with probability 0.5. For ICDAR, we trained each network on a single Tesla K20m GPU for a total 45 epochs with a batch size of 256 and 1,000 iterations per epoch. In this case, the learning rate was set to an initial value of 0.1 and multiplied by 0.1 in the second, seventh and fifteenth epochs. We used a momentum of 0.9. In terms of hyper-parameters, for large-scale classification, we used $\lambda_l = 0.102$ for the first three layers and $\lambda_l = 0.255$ for the remaining ones. For ICDAR, we used $\lambda_l = 5.1$ for the first layer and $\lambda_l = 10.2$ for the remaining ones.

**Evaluation:** We measure classification performance as the top-1 accuracy using the center crop, referred to as **Top-1**. We compare the results of our approach with those obtained by training the same architectures, but without our model selection technique. We also provide the results of additional, standard architectures. Furthermore, since our approach can determine the number of neurons per layer, we also computed results with our method starting for different number of neurons, referred to as $M$ below, in the overcomplete network. In addition to accuracy, we also report, for the convolutional layers, the percentage of neurons set to 0 by our approach (**neurons**), the corresponding percentage of zero-valued parameters (**group param**), the total percentage of 0 parameters (**total param**), which additionally includes the parameters set to 0 in non-completely zeroed-out neurons, and the total percentage of zero-valued parameters induced by the zeroed-out neurons (**total induced**), which additionally includes the neurons in each layer, including the last fully-connected layer, that have been rendered useless by the zeroed-out neurons of the previous layer.

## 4.2 Results

Below, we report our results on ImageNet and ICDAR. The results on Places-2 are provided as supplementary material.

**ImageNet:** We first start by discussing our results on ImageNet. For this experiment, we used BNet$^C$ and Dec$_8$, both with the group sparsity (GS) regularizer of Eq. 2. Furthermore, in the case of

Table 1: Top-1 accuracy results for several state-of-the art architectures and our method on ImageNet.

| Model | Top-1 acc. (%) |
|---|---|
| BNet | 62.5 |
| $BNet^C$ | 61.1 |
| $ResNet50^a$ [He et al., 2015] | 67.3 |
| $Dec_8$ | 64.8 |
| $Dec_8$-640 | 66.9 |
| $Dec_8$-768 | 68.1 |

| Model | Top-1 acc. (%) |
|---|---|
| $Ours\text{-}Bnet_{GS}^C$ | 62.7 |
| $Ours\text{-}Dec_{8-GS}$ | 64.8 |
| $Ours\text{-}Dec_8\text{-}640_{SGL}$ | 67.5 |
| $Ours\text{-}Dec_8\text{-}640_{GS}$ | 68.6 |
| $Ours\text{-}Dec_8\text{-}768_{GS}$ | 68.0 |

$^a$ Trained over 55 epochs using a batch size of 128 on two TitanX with code publicly available.

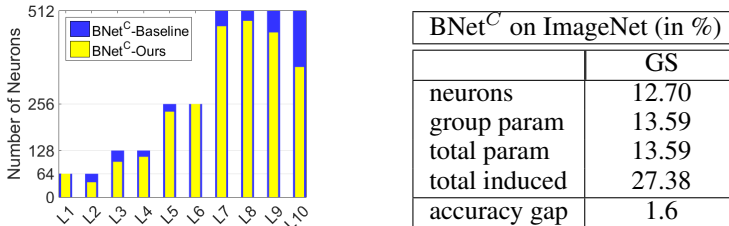

| $BNet^C$ on ImageNet (in %) | |
|---|---|
| | GS |
| neurons | 12.70 |
| group param | 13.59 |
| total param | 13.59 |
| total induced | 27.38 |
| accuracy gap | 1.6 |

Figure 1: Parameter reduction on ImageNet using $BNet^C$. (Left) Comparison of the number of neurons per layer of the original network with that obtained using our approach. (Right) Percentage of zeroed-out neurons and parameters, and accuracy gap between our network and the original one. Note that we outperform the original network while requiring much fewer parameters.

$Dec_8$, we evaluated two additional versions that, instead of the 512 neurons per layer of the original architecture, have $M = 640$ and $M = 768$ neurons per layer, respectively. Finally, in the case of $M = 640$, we further evaluated both the group sparsity regularizer of Eq. 2 and the sparse group Lasso (SGL) regularizer of Eq. 3 with $\alpha = 0.5$. Table 1 compares the top-1 accuracy of our approach with that of the original architectures and of other baselines. Note that, with the exception of $Dec_8$-768, all our methods yield an improvement over the original network, with up to 1.6% difference for $BNet^C$ and 2.45% for $Dec_8$-640. As an additional baseline, we also evaluated the naive approach consisting of reducing each layer in the model by a constant factor of 25%. The corresponding two instances, $Dec_8^{25\%}$ and $Dec_8^{25\%}$-640, yield 64.5% and 65.8% accuracy, respectively.

More importantly, in Figure 1 and Figure 2, we report the relative saving obtained with our approach in terms of percentage of zeroed-out neurons/parameters for $BNet^C$ and $Dec_8$, respectively. For $BNet^C$, in Figure 1, our approach reduces the number of neurons by over 12%, while improving its generalization ability, as indicated by the accuracy gap in the bottom row of the table. As can be seen from the bar-plot, the reduction in the number of neurons is spread all over the layers with the largest difference in the last layer. As a direct consequence, the number of neurons in the subsequent fully connected layer is significantly reduced, leading to 27% reduction in the total number of parameters. For $Dec_8$, in Figure 2, we can see that, when considering the original architecture with 512 neurons per layer, our approach only yields a small reduction in parameter numbers with minimal gain in performance. However, when we increase the initial number of neurons in each layer, the benefits of our approach become more significant. For $M = 640$, when using the group sparsity regularizer, we see a reduction of the number of parameters of more than 19%, with improved generalization ability. The reduction is even larger, 23%, when using the sparse group Lasso regularizer. In the case of $M = 768$, we managed to remove 26% of the neurons, which translates to 48% of the parameters. While, here, the accuracy is slightly lower than that of the initial network, it is in fact higher than that of the original $Dec_8$ network, as can be seen in Table 1.

Interestingly, during learning, we also noticed a significant reduction in the training-validation accuracy gap when applying our regularization technique. For instance, for $Dec_8$-768, which zeroes out 48.2% of the parameters, we found the training-validation gap to be 28.5% smaller than in the original network (from 14% to 10%). We believe that this indicates that networks trained using our approach have better generalization ability, even if they have fewer parameters. A similar phenomenon was also observed for the other architectures used in our experiments.

We now analyze the sensitivity of our method with respect to $\lambda_l$ (see Eq. (2)). To this end, we considered $Dec_8 - 768_{GS}$ and varied the value of the parameter in the range $\lambda_l = [0.051..0.51]$. More specifically, we considered 20 different pairs of values, $(\lambda^1, \lambda^2)$, with the former applied to the

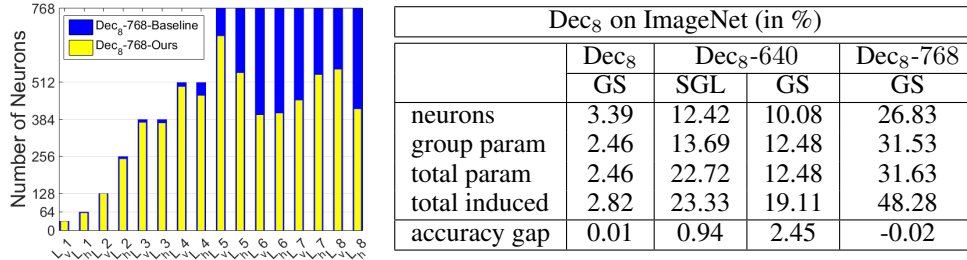

| Dec$_8$ on ImageNet (in %) | | | |
|---|---|---|---|
| | Dec$_8$ | Dec$_8$-640 | Dec$_8$-768 |
| | GS | SGL \| GS | GS |
| neurons | 3.39 | 12.42 \| 10.08 | 26.83 |
| group param | 2.46 | 13.69 \| 12.48 | 31.53 |
| total param | 2.46 | 22.72 \| 12.48 | 31.63 |
| total induced | 2.82 | 23.33 \| 19.11 | 48.28 |
| accuracy gap | 0.01 | 0.94 \| 2.45 | -0.02 |

Figure 2: Parameter reduction using Dec$_8$ on ImageNet. Note that we significantly reduce the number of parameters and, in almost all cases, improve recognition accuracy over the original network.

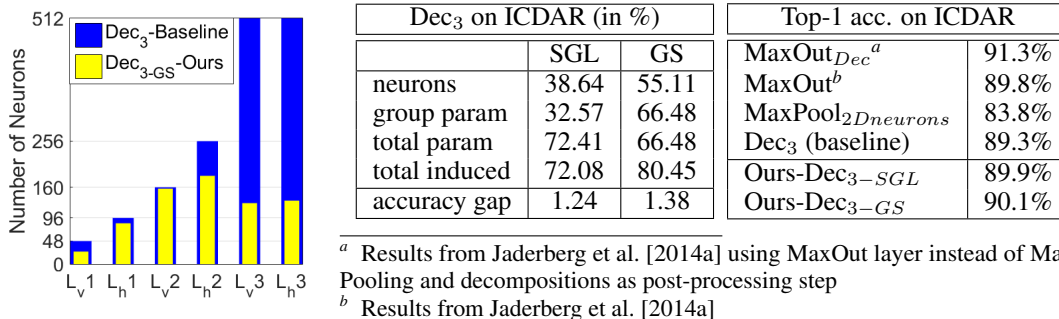

| Dec$_3$ on ICDAR (in %) | | | Top-1 acc. on ICDAR | |
|---|---|---|---|---|
| | SGL | GS | MaxOut$_{Dec}$[a] | 91.3% |
| neurons | 38.64 | 55.11 | MaxOut[b] | 89.8% |
| group param | 32.57 | 66.48 | MaxPool$_{2Dneurons}$ | 83.8% |
| total param | 72.41 | 66.48 | Dec$_3$ (baseline) | 89.3% |
| total induced | 72.08 | 80.45 | Ours-Dec$_{3-SGL}$ | 89.9% |
| accuracy gap | 1.24 | 1.38 | Ours-Dec$_{3-GS}$ | 90.1% |

[a] Results from Jaderberg et al. [2014a] using MaxOut layer instead of Max-Pooling and decompositions as post-processing step
[b] Results from Jaderberg et al. [2014a]

Figure 3: Experimental results on ICDAR using Dec$_3$. Note that our approach reduces the number of parameters by 72% while improving the accuracy of the original network.

first three layers and the latter to the remaining ones. The details of this experiment are reported in supplementary material. Altogether, we only observed small variations in validation accuracy (std of 0.33%) and in number of zeroed-out neurons (std of 1.1%).

**ICDAR:** Finally, we evaluate our approach on a smaller dataset where architectures have not yet been heavily tuned. For this dataset, we used the Dec$_3$ architecture, where the last two layers initially contain 512 neurons. Our goal here is to obtain an optimal architecture for this dataset. Figure 3 summarizes our results using GS and SGL regularization and compares them to state-of-the-art baselines. From the comparison between MaxPool$_{2Dneurons}$ and Dec$_3$, we can see that learning 1D filters leads to better performance than an equivalent network with 2D kernels. More importantly, our algorithm reduces by up to 80% the number of parameters, while further improving the performance of the original network. We believe that these results evidence that our algorithm effectively performs automatic model selection for a given (classification) task.

### 4.3 Benefits at test time

We now discuss the benefits of our algorithm at test time. For simplicity, our implementation does not remove neurons during training. However, these neurons can be effectively removed after training, thus yielding a smaller network to deploy at test time. Not only does this entail benefits in terms of memory requirement, as illustrated above when looking at the reduction in number of parameters, but it also leads to speedups compared to the complete network. To demonstrate this, in Table 2, we report the relative runtime speedups obtained by removing the zeroed-out neurons. For BNet and Dec$_8$, these speedups were obtained using ImageNet, while Dec$_3$ was tested on ICDAR. Note that significant speedups can be achieved, depending on the architecture. For instance, using BNet$^C$, we achieve a speedup of up to 13% on ImageNet, while with Dec$_3$ on ICDAR the speedup reaches almost 50%. The right-hand side of Table 2 shows the relative memory saving of our networks. These numbers were computed from the actual memory requirements in MB of the networks. In terms of parameters, for ImageNet, Dec$_8$-768 yields a 46% reduction, while Dec$_3$ increases this saving to more than 80%. When looking at the actual features computed in each layer of the network, we reach a 10% memory saving for Dec$_8$-768 and a 25% saving for Dec$_3$. We believe that these numbers clearly evidence the benefits of our approach in terms of speed and memory footprint at test time.

Note also that, once the models are trained, additional parameters can be pruned using, at the level of individual parameters, $\ell_1$ regularization and a threshold [Liu et al., 2015]. On ImageNet, with our

Table 2: Gain in runtime (actual clock time) and memory requirement of our reduced networks. Note that, for some configurations, our final networks achieve a speedup of close to 50%. Similarly, we achieve memory savings of up to 82% in terms of parameters, and up to 25% in terms of the features computed by the network. The runtimes were obtained using a single Tesla K20m and memory estimations using RGB-images of size $224 \times 224$ for Ours-BNet$^C$, Ours-Dec$_8$-640$_{GS}$ and Ours-Dec$_8$-768$_{GS}$, and gray level images of size $32 \times 32$ for Ours-Dec$_{3-GS}$.

| | Relative speed-up (%) | | | | Relative memory-savings | |
| | Batch size | | | | Batch size 1 (%) | |
| Model | 1 | 2 | 8 | 16 | Params | Features |
|---|---|---|---|---|---|---|
| Ours-BNet$^C_{GS}$ | 10.04 | 8.97 | 13.01 | 13.69 | 12.06 | 18.54 |
| Ours-Dec$_8$-640$_{GS}$ | -0.1 | 5.44 | 3.91 | 4.37 | 26.51 | 2.13 |
| Ours-Dec$_8$-768$_{GS}$ | 15.29 | 17.11 | 15.99 | 15.62 | 46.73 | 10.00 |
| Ours-Dec$_{3-GS}$ | 35.62 | 43.07 | 44.40 | 49.63 | 82.35 | 25.00 |

Dec$_8$-768$_{GS}$ model and the $\ell_1$ weight set to 0.0001 as in [Liu et al., 2015], this method yields 1.34M zero-valued parameters, compared to 7.74M for our approach, i.e., a $82\%$ relative reduction in the number of individual parameters for our approach.

## 5    Conclusions

We have introduced an approach to automatically determining the number of neurons in each layer of a deep network. To this end, we have proposed to rely on a group sparsity regularizer, which has allowed us to jointly learn the number of neurons and the parameter values in a single, coherent framework. Not only does our approach estimate the number of neurons, it also yields a more compact architecture than the initial overcomplete network, thus saving both memory and computation at test time. Our experiments have demonstrated the benefits of our method, as well as its generalizability to different architectures. One current limitation of our approach is that the number of layers in the network remains fixed. To address this, in the future, we intend to study architectures where each layer can potentially be bypassed entirely, thus ultimately canceling out its influence. Furthermore, we plan to evaluate the behavior of our approach on other types of problems, such as regression networks and autoencoders.

**Acknowledgments**

The authors thank John Taylor and Tim Ho for helpful discussions and their continuous support through using the CSIRO high-performance computing facilities. The authors also thank NVIDIA for generous hardware donations.

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
