[Reviews · NeurIPS 2016]

Reviewer 1

Summary

Use group sparse lasso (with SGD) to prune neurons in deep networks applied to visual problems. Reduces neurons by ~20% on various problems, while often increasing accuracy a bit (~1%).

Qualitative Assessment

Overall, a nice paper. I lean towards acceptance. A few questions/comments: 1. In section 4.3, are the test-time speedups computed via counting multiply-adds, or actual clock time? For sparse models, it's very important to do measure and report actual clock time, because that is subtle function of how the sparse matrices are structured. 2. The authors missed an important prior piece of work in CVPR 2015, see reference [1]. That paper trains a network, performs tensor decomposition on each layer, and then does fine tuning with group lasso on the tensor blending weights. They report saving ~90% parameters in a CNN, vs current submission saving of ~20%. However, the other paper doesn't describe how they train with the group lasso --- using the proximal method is a sensible method, although fairly obvious at this point. Critically, the other paper doesn't compare to applying group lasso by itself (without tensor decomposition). They also report an accuracy loss (rather than gain). So, I think that this paper still deserves to be published, because it fills in a gap in our knowledge about group lasso and neural networks. 3. Incrementally growing neural networks dates back to 1989, see [2]. References: 1. Liu, B., Wang, M., Foroosh, H., Tappen, M., & Pensky, M. (2015). Sparse convolutional neural networks. In Proceedings of the IEEE Conference on Computer Vision and Pattern Recognition (pp. 806-814). http://www.cv-foundation.org/openaccess/content_cvpr_2015/html/Liu_Sparse_Convolutional_Neural_2015_CVPR_paper.html 2. Ash, Timur. "Dynamic node creation in backpropagation networks." Connection Science 1.4 (1989): 365-375. http://www.cogsci.ucsd.edu/research/documents/technical/TR-8901.pdf

Confidence in this Review

3-Expert (read the paper in detail, know the area, quite certain of my opinion)


Reviewer 2

Summary

The paper applies group sparsity (L1 and L2) on weights into each node in a conv net. They show small improvements in accuracy and useful reductions in the number of nodes relative to baseline models. There is no comparison to alternative methods of sparsity induction such as the practice of training and then clamping small weights to zero (e.g. the microsoft paper cited below,and Han cited by the recent arxiv paper)

Qualitative Assessment

idea of group sparsity for neural network weights doesn’t seem new (though I don’t have a citation) how much does it slow down training? accuracy gap is not clear - that positive means that you improved the accuracy how about “gain”? State earlier that baseline Dec8 has 512 neurons. Better just put Dec_3_512 everywhere Comparing Dec_3 768 you say “initial” network and “original” network which are a bit ambiguous. That your pruned 768 network should be more accurate than the 'original dec_8’ is hardly surprising You should compare to conventional L1 and L2 regularization. which can reduce the number of non-zero parameters, as well as other techniques such as thresholding small weights, and clamping them to zero while further fine tuning. see e.g. https://www.microsoft.com/en-us/research/publication/exploiting-sparseness-in-deep-neural-networks-for-large-vocabulary-speech-recognition/ as well as optimal brain damage and related techniques Figure 4: label as reduction in number of neurons. There is no explanation of the effect of different lambda parameters (which vary by two orders of magnitude in your chosen values, yet , suspiciously, are quoted to 3 significant figures…) , nor the extent to which you explored varying them. Please include this in any revision. You should cite the simultaneous http://arxiv.org/pdf/1607.00485

Confidence in this Review

2-Confident (read it all; understood it all reasonably well)


Reviewer 3

Summary

The paper introduces a group-sparsity regularization acting on single units in the network to favor solutions that use only a small subset of the original parameters. The resulting model can be made much smaller and therefore more efficient to compute.

Qualitative Assessment

The paper is clearly written, I find the title too bold in my opinion though. Model selection can refer to a much broader topic, I would keep it more to the point of the paper. For example when reading the title I thought it was about selecting what architecture choices were best for a given task. Instead the paper presents a way of pruning away blocks of parameters (thus removing a unit) from a given overly parameterised fixed network. In a fully connected layer if one neuron is always off then you can remove it. This means that the set of weights used to compute it, the entire column (or row) can be discarded therefore mimicking the group sparsity outcome. What is the relationship between these two penalty terms? The idea is nevertheless interesting and has many potential practical implications. Experiments are performed on a large variety of datasets and well conducted. However, I think that the following important comparisons are missing: - L1 penalty + thresholding on the units (simple and brute force) - Distilling the knowledge in a neural network. This latter I think is important because in the abstract the authors mention resource constrained platforms, which are indeed of interest in industry. I'd like to see, given a certain upper bound on the resources, which algorithm performs best. The approach of Hinton and colleagues does no model selection but allows to optimize for a topology which can be mapped to a particular device, whereas the proposed approach lacks such flexibility. How about convergence times? Is the group-sparsity regularization helping? Source code should be made available in case of publication.

Confidence in this Review

2-Confident (read it all; understood it all reasonably well)


Reviewer 4

Summary

This paper proposes a novel regulaizer that can be applide to convolutional networks to automatically select the size of each convolutional layer. The regulaizer itself is a combination of L1 and L2 regularizers. The approach is a reasonable enough, and the only major reservation i have is about experimental part of the paper.

Qualitative Assessment

The approach is quite reasonable, however my biggest concern for this paper is lack of any baseline evaluation. That is: how this approach compares to naive approach, where say every layer just reduced by a constant factor (e.g. say 25%). This would be a very valuable sanity check for this metric. Also, there is related work available on auto-sizing entworks, which seems to employ very similar approach: http://arxiv.org/pdf/1508.05051v1.pdf, it would be great if some discussion was included.

Confidence in this Review

2-Confident (read it all; understood it all reasonably well)


Reviewer 5

Summary

Group lasso regularization was added to remove neurons for deep model selection. Reported experimental results on a couple of deep models showed the expected model reduction with comparable prediction accuracy on several benchmark datasets.

Qualitative Assessment

Stochastic proximal gradient descent was presented to solve the deep model training with the proposed model selection based on group lasso regularization. However, the implementation details were not presented, especially regarding the iterative updates for the "loss-based gradient step". From the context, it seems that the iterative updates in the original deep model training procedure were simply modified based on Eq. 5. If that is the case, the claim of the generality of the method for "general deep networks" as it will depend on the optimization procedure of the deep model training. If the motivation of the presented work is to remove redundant neurons. it appears natural to compare the proposed method with the dropout method to see the performance. Without that, it is difficult to evaluate the significance and advantages of the method. The presented performance evaluation is problematic. It is somewhat strange that both the performance evaluation results were done based on some modifications of the original deep models, either removing the first two layers (BnetC) or changing maxout to maxpooling (Dec3). Why was the comparison not done to the original model without changes, especially considering the reported decrease of top-1 accuracy in Table 1? It is not described how "accuracy gap" was computed. If it is simply the top-1 accuracy difference, it does not seem to be consistent based on the tables in Figs 3 and 4. More critically, as typically done in sparse models, the sensitivity of the method with respect to penalty coefficients should be analyzed, especially with the complicated loss function in deep models. There are also many confusing places in experiments. For example, in Table 1, what is the difference between BnetC and "Ours-BnetC" and for other similar pairs? What does "S-GS" stand for (SGL?) in the table of Fig. 4? There are also typos throughout the paper. For example, in line 275: "zeroed-our" should be "zeroed-out"; in line 293, "were" should be "where"?

Confidence in this Review

2-Confident (read it all; understood it all reasonably well)